# Comparative Gene Expression Analysis Reveals Similarities and Differences of Chronic Myeloid Leukemia Phases

**DOI:** 10.3390/cancers14010256

**Published:** 2022-01-05

**Authors:** Annemarie Schwarz, Ingo Roeder, Michael Seifert

**Affiliations:** 1Institute for Medical Informatics and Biometry (IMB), Carl Gustav Carus Faculty of Medicine, Technische Universität Dresden, D-01307 Dresden, Germany; annemarie.schwarz@mailbox.tu-dresden.de (A.S.); ingo.roeder@tu-dresden.de (I.R.); 2National Center for Tumor Diseases (NCT), D-01307 Dresden, Germany: German Cancer Research Center (DKFZ), D-69120 Heidelberg, Germany; Faculty of Medicine and University Hospital Carl Gustav Carus, Technische Universität Dresden, D-01307 Dresden, Germany; Helmholtz-Zentrum Dresden—Rossendorf (HZDR), D-01328 Dresden, Germany

**Keywords:** chronic myeloid leukemia (CML), computational cancer genomics, comparative transcriptome analysis, chronic phase, accelerated phase, blast crisis

## Abstract

**Simple Summary:**

Chronic myeloid leukemia (CML) is a blood cancer with a very good long-term prognosis for the majority of patients. Still, some patients relapse or progress under the standard therapy. Therefore, we performed an in-depth computational gene expression analysis to determine similarities and differences between the CML phases at the level of single genes, signaling pathways and gene regulatory networks and further compared treatment-resistant patients to the individual phases to identify associated expression differences. Our study provides several lines of evidence that CML development represents a three rather than a two step process. The identified characteristic gene expression alterations in advanced phases indicate that affected patients could potentially profit from other existing drugs. Moreover, characteristic signaling pathway changes identified for patients for which the standard therapy was inefficient may allow to predict such patients before treatment start and could also provide a basis to develop treatment strategies for them.

**Abstract:**

Chronic myeloid leukemia (CML) is a slowly progressing blood cancer that primarily affects elderly people. Without successful treatment, CML progressively develops from the chronic phase through the accelerated phase to the blast crisis, and ultimately to death. Nowadays, the availability of targeted tyrosine kinase inhibitor (TKI) therapies has led to long-term disease control for the vast majority of patients. Nevertheless, there are still patients that do not respond well enough to TKI therapies and available targeted therapies are also less efficient for patients in accelerated phase or blast crises. Thus, a more detailed characterization of molecular alterations that distinguish the different CML phases is still very important. We performed an in-depth bioinformatics analysis of publicly available gene expression profiles of the three CML phases. Pairwise comparisons revealed many differentially expressed genes that formed a characteristic gene expression signature, which clearly distinguished the three CML phases. Signaling pathway expression patterns were very similar between the three phases but differed strongly in the number of affected genes, which increased with the phase. Still, significant alterations of MAPK, VEGF, PI3K-Akt, adherens junction and cytokine receptor interaction signaling distinguished specific phases. Our study also suggests that one can consider the phase-wise CML development as a three rather than a two-step process. This is in accordance with the phase-specific expression behavior of 24 potential major regulators that we predicted by a network-based approach. Several of these genes are known to be involved in the accumulation of additional mutations, alterations of immune responses, deregulation of signaling pathways or may have an impact on treatment response and survival. Importantly, some of these genes have already been reported in relation to CML (e.g., *AURKB*, *AZU1*, *HLA-B*, *HLA-DMB*, *PF4*) and others have been found to play important roles in different leukemias (e.g., *CDCA3*, *RPL18A*, *PRG3*, *TLX3*). In addition, increased expression of *BCL2* in the accelerated and blast phase indicates that venetoclax could be a potential treatment option. Moreover, a characteristic signaling pathway signature with increased expression of cytokine and ECM receptor interaction pathway genes distinguished imatinib-resistant patients from each individual CML phase. Overall, our comparative analysis contributes to an in-depth molecular characterization of similarities and differences of the CML phases and provides hints for the identification of patients that may not profit from an imatinib therapy, which could support the development of additional treatment strategies.

## 1. Introduction

Chronic myeloid leukemia (CML) is a myeloproliferative neoplasm that accounts for about 15% of all leukemias diagnosed in adulthood [1]. About one to two cases occur per year among 100,000 adults [1]. CML predominately occurs in elderly people with a median age between 57 and 60 years at diagnosis in Europe [2].

As described in [3], CML development is initiated in about 95% of cases by a characteristic reciprocal translocation between the chromosomes 9 and 22 (t(9;22)(q34.1;q11.2)), which produces a shortened chromosome 22 that is known as Philadelphia chromosome. The breakpoints of this translocation are located within the breakpoint cluster region (*BCR*) gene on chromosome 22 and the Abelson murine leukemia viral oncogene homolog 1 (*ABL1*) gene on chromosome 9 leading to the formation of the *BCR-ABL1* fusion gene on the shortened chromosome 22. The resulting *BCR-ABL1* fusion gene is an oncogene that encodes a constitutively active tyrosine kinase, which triggers uncontrolled cell proliferation. This translocation can transform a hematopoietic stem or progenitor cell into a leukemic cell in which the expression of the *BCR-ABL1* fusion gene leads to a continuous clonal expansion of leukemic cells. Normal blood cells are progressively outcompeted by these leukemic cells due to their clonal expansion resulting in the manifestation of CML in a chronic phase characterized by myeloid hyperplasia and indolent symptoms.

Without successful treatment, CML develops from the chronic phase through an accelerated phase into a blast crisis, which resembles an acute leukemia [3]. The strong enrichment of immature blood cells in the bone marrow and the peripheral blood during the blast crisis severely limits the development of normal blood cells leading almost always to a fatal outcome.

With the approval of the tyrosine kinase inhibitor imatinib in 2001 as the first targeted drug therapy that directly blocks the activity of the BCR-ABL1 oncokinase [4], the progression of CML patients to the blast crisis has been substantially reduced and continuous treatment with imatinib has enabled long-term disease control, where most CML patients remain in remission [1,5,6]. The survival rate of CML patients after 10 years of imatinib treatment has been reported to be greater than 83% [6]. Similarly, a survival rate of 88.3% after 10 years of imatinib treatment has been reported in [7]. In addition, the availability of second- (dasatinib, nilotinib) and third-generation tyrosine kinase inhibitors (bosutinib, ponatinib) further improves therapy efficacy and provides additional targeted treatment strategies to overcome resistance against imatinib [8,9].

Due to the excellent treatment responses to available targeted treatment strategies, a development from the chronic phase to the blast crisis is only rarely observed today [6,10]. Nevertheless, in some cases CML is still first diagnosed at the stage of blast crisis [11,12,13] and in other cases the gain of additional mutations can lead to treatment failure and CML progression [8]. Unfortunately, high and long-term response rates to targeted tyrosine kinase inhibitors are mainly achieved for the chronic phase, but such treatments are less efficient in accelerated phase or blast crisis [14,15,16]. Thus, a more detailed understanding of alterations that distinguish the different CML phases is required.

Over the last years, CML has been analyzed at the molecular level in many different studies. An important contribution to this was the identification of progression-associated gene expression changes by [17] based on genome-wide CML expression profiles of all three phases on the basis of phase-specific samples from different patients. Other studies focused on the comparison of CML cells to normal cells revealing altered cell properties and signaling pathway alterations in the background of the *BCR-ABL1* driver mutation (e.g., [18,19,20]). In addition, a recent study has performed an integrative multi-omics analysis to better understand differences between the chronic phase and the blast crisis [21]. Another study compared TKI-resistant and -sensitive CML samples predicting altered genes, signaling pathways and additional mutations that may allow to monitor therapy response [22]. Further, entropy-based modeling of CML expression profiles has suggested a separation into an early and late chronic phase [23]. Despite these different advances, additional studies are still necessary to better characterize molecular changes that distinguish the three CML phases. Such knowledge is essential to provide a basis for the development of improved treatment strategies for advanced CML states to improve the therapy success for patients that do not benefit from highly efficient tyrosine kinase inhibitor therapies.

Here, we present an in-depth comparative bioinformatics reanalysis of CML gene expression profiles from [17] to contribute to a better characterization of similarities and differences between the three CML phases at the level of single genes, signaling pathways and gene regulatory networks. In addition, we also analyze gene expression profiles of patients that were insensitive to imatinib treatment to search for molecular patterns that are in accordance with or distinguish them from other patients of each of the three CML phases.

## 2. Results

### 2.1. Global Gene Expression Characterization of CML Samples

Genome-wide gene expression profiles of 87 CML samples (chronic phase: 42, accelerated phase: 17, blast crisis: 28) from [17] were analyzed by a principal component analysis (PCA) enabling an exploratory analysis of the distribution of the samples from the three different CML phases in two dimensions (Figure 1). This PCA analysis provides an initial overview about the similarity of individual samples in relation to each other and further allows to see if the samples of the different phases tend to form separate clusters. We found that chronic phase samples were clearly separated from blast crisis samples. Accelerated phase samples tended to lie in between chronic phase and blast crisis samples. Further, chronic phase samples were more heterogeneous than blast crisis samples. Samples of the accelerated phase that were classified by blast counts were more homogeneous than accelerated phase samples that were assigned to this class based on additional cytogenetic alterations. Generally, the phase-wise development of CML known from histology is also reflected in the global gene expression profiles of CML samples at the molecular level.

### 2.2. Differential Gene Expression Analysis of CML Phases

Motivated by the phase-wise clustering of the three different CML phases (Figure 1), a detailed analysis of individual gene expression differences was performed by pairwise comparisons of the CML phases (Figure 2, Appendix A). This differential gene expression analysis enabled to predict individual genes that differ in their average expression between two phases. The pairwise comparisons of the three different CML phases showed that expression differences between the accelerated and the chronic phase were clearly smaller than those between the blast and the accelerated phase in terms of the number of differentially expressed genes and the strength of expression differences (Figure 2a,b, *q*-value ≤ 0.05, accelerated vs. chronic: 1960 genes, blast vs. accelerated: 5437 genes). The largest expression differences were observed between the blast and the chronic phase (Figure 2c, *q*-value ≤ 0.05: 8196 genes). Further, the number of underexpressed genes increased with the phase and was greater than the corresponding increase of overexpressed genes (Figure 2, Appendix A, *q*-value ≤ 0.05: accelerated vs. chronic: 1101 under- and 859 overexpressed genes, blast vs. accelerated: 2823 under- and 2614 overexpressed genes, blast vs. chronic: 4384 under- and 3812 overexpressed genes). These numbers were clearly reduced when focusing only on differentially expressed genes with stronger expression differences between two phases (Figure 2, Appendix A, *q*-value ≤ 0.05 and average absolute expression change |log${}_{2}$-ratio| ≥ 1: accelerated vs. chronic: 111 under- and 312 overexpressed genes, blast vs. accelerated: 664 under- vs. 416 overexpressed genes, blast vs. chronic: 827 under- vs. 891 overexpressed genes).

An analysis of general functional annotation categories of all differentially expressed genes revealed that many of these genes encode for transcription factors or co-factors, but also other functional categories such as signaling or metabolic pathway genes or known cancer census genes were affected by expression changes comparing the three CML phases (Figure 2). In more detail, several of these functional categories were significantly enriched for genes (FDR-adjusted *p*-values ≤ 0.05, accelerated vs. chronic: phosphatases; blast vs. accelerated: phosphatases, metabolic genes, cancer census genes, essential genes; blast vs. chronic: kinases, phosphatases, signaling genes). This included for example the three well-known tumor suppressor genes *APC*, *CDKN2C* and *TGFBR2*, which were underexpressed in blast crisis in comparison to chronic phase, and the four well-known oncogenes *BCL2*, *ERBB2*, *HRAS* and *NRAS*, which were overexpressed in blast crisis compared to chronic phase. In addition, *ERBB2* was overexpressed in the accelerated compared to the chronic phase. Further, *BCL2* and *HRAS* were overexpressed in the blast compared to the accelerated phase, whereas *APC* and *TGFBR2* were underexpressed in this context.

### 2.3. Alterations of Cancer-Relevant Signaling Pathways Increase with CML Phase

Under- and overexpressed genes identified at the *q*-value cutoff of 0.05 were further considered for an in-depth analysis of transcriptional alterations of specific cancer-relevant signaling pathways (Figure 3). This analysis of differentially expressed genes at the level of signaling pathways enables to analyze the expression behavior of individual pathways and allows to characterize similarities and differences between the three CML phases. In accordance with the differential gene expression analysis, more alterations of signaling pathway genes were observed for the comparison of the blast to the accelerated phase than for the comparison of the accelerated to the chronic phase (Figure 3a,b). Most alterations of signaling pathways were observed for the comparison of the blast to the chronic phase (Figure 3c). Generally, more underexpression than overexpression of signaling pathway genes was observed. All included signaling pathways contained genes with altered expression for the pairwise comparison of the three CML phases, except for some DNA repair pathways in the comparison of the accelerated to the chronic phase. The global pattern of affected signaling pathways did not strongly differ between the pairwise comparisons of the three CML phases, but the number of affected genes successively increased from the comparison of the accelerated to the chronic phase over the comparison of the blast to the accelerated phase up to the comparison of the blast to the chronic phase (Figure 3). Several signaling pathways were significantly enriched for underexpressed genes (Figure 3, FDR-adjusted *p*-value ≤ 0.05, accelerated vs. chronic: cytokine receptor interaction; blast vs. accelerated: MAPK signaling, adherens junction; blast vs. chronic: MAPK signaling, PI3K-Akt signaling, VEGF signaling), but no enrichment of overexpressed genes was observed for an individual pathway for the comparisons of the three CML phases. Focusing on differentially expressed genes with stronger expression differences between the CML phases (Appendix A, *q*-value ≤ 0.05 and average absolute expression change |log${}_{2}$-ratio| ≥ 1), only the cytokine receptor interaction pathway was enriched for differentially expressed genes. This pathway was enriched for overexpressed genes for the comparison of the accelerated to the chronic phase, whereas an enrichment of underexpressed genes was observed for the comparisons of the blast to the accelerated and of the blast to the chronic phase (Appendix A).

### 2.4. Global Expression Signature Distinguishes CML Phases

The pairwise comparisons of the expression levels of individual genes of the three CML phases involved a large number of samples and especially chronic and accelerated phase samples tended to be more heterogeneous than blast crisis samples (Figure 1). In such a situation, it is possible that an individual gene, which was predicted to be differentially expressed between two CML phases, still varies to some extent in its expression across the samples of a specific phase. Therefore, all 9884 differentially expressed genes predicted in the pairwise comparisons of the three different CML phases at a *q*-value cutoff of 0.05 were used to perform a hierarchical clustering of all individual CML samples to analyze how well these genes can separate the three CML phases in general (Figure 4, Appendix A).

This clustering almost perfectly separated chronic phase samples from accelerated and blast samples that were part of one joint cluster (Figure 4: left cluster: accelerated (red) and blast samples (light blue), right cluster: chronic samples (yellow)). The chronic phase cluster was further divided into three larger subclusters and also only contained two accelerated phase samples that had expression profiles which were highly similar to those of chronic samples. The other cluster, which contained all other accelerated phase samples and all blast phase samples, was further separated into two larger subclusters. One of these subclusters contained almost all blast phase samples, whereas the other subcluster contained the majority of accelerated phase samples. Globally, the expression profiles of accelerated phase samples were much more similar to blast phase samples than to chronic phase samples. Nevertheless, differences between accelerated and blast phase were clearly visible supporting that the accelerated phase is a transition phase between the chronic phase and the blast crisis.

### 2.5. CML Signature-Specific Gene Regulatory Network

The 9884 differentially expressed genes predicted by the pairwise comparisons of the three CML phases (Figure 4, Appendix A) were further utilized to learn gene regulatory networks to predict potential major regulators that differ in their expression between the CML phases. This was done using the R package regNet [24] (see Materials and Methods section for details). The basic idea behind this approach is to predict the expression behavior of a gene based on the expression levels of other genes that can best explain the observed expression behavior of the specific gene across the three CML phases. This is repeated 100 times based on randomly chosen subsets of all CML samples to only focus on links between genes that are found in the majority of the different network inference runs. Overall, the 100 learned networks contained relevant information to predict the expression behavior of genes in CML significantly better than corresponding random networks of same complexity (Appendix A, median correlations: 0.63 vs. 0.01, U-test: *p* < $2.2\times 10^{-16}$). These networks were used to create a consensus network by focusing on robust links between genes that were present in at least 90 of 100 networks (link cutoff: *q*≤ 0.01). All genes with at least two outgoing links to other genes are included in Figure 5. The left network (Figure 5a) shows the expression differences of these genes for the comparison of the accelerated to the chronic phase and those for the comparison of the blast to accelerated phase are shown in the corresponding right network (Figure 5b). Since both networks have the same topological structure, these network representations additionally highlight expression differences of phase-specific alterations from the chronic to the blast phase.

This basic network structure also contained 24 genes that had outgoing links to at least three other genes (Table 1). These genes differed in their expression behavior and can be divided into three general groups: (i) genes whose expression was reduced towards the blast crisis (*ADD2*, *AURKB*, *AZU1*, *CDCA3*, *CEACAM6*, *CTRB1*, *ECEL1*, *HLA-B*, *INMT*, *PRG3*), (ii) genes whose expression was increased towards the blast crisis (*HLA-DMB*, *HLA-DRA*, *NDUFAB1*, *OPTN*), and (iii) genes that distinguished the accelerated phase from the chronic phase and the blast crisis (*EN1*, *LOC284023*, *LOC389458*, *SPRR2A*, *MUC8*, *PF4*, *RHBDL1*, *RPL18A*, *TLX3*, *TMEM40*).

Condensed results of an in-depth gene annotation analysis [25] of these genes are briefly summarized (Table 1). As one may expect, several of these genes are involved in the regulation of cell proliferation, adhesion, apoptosis, migration, differentiation, or invasion (*ADD2*, *AURKB*, *CDCA3*, *CEACAM6*, *TMEM40*). Others are involved in the regulation of immune responses and inflammation processes (*HLA-DMB*, *HLA-DRA*, *HLA-B*, *MUC8*, *OPTN*). Two encode for homeobox transcription factors (*EN1*, *TLX3*), and two other genes are involved in oxidative stress regulation (*NDUFAB1*, *SPRR2A*). Further, an in-depth literature analysis revealed that several of these genes play important roles in cancer (*CDCA3* [26], *ECEL1* [27], *EN1* [28], *HLA-DMB* [29,30], *SPRR2A* [31], *TMEM40* [32,33,34]) including acute myeloid leukemia (*CDCA3* [35], *HLA-DMB* [30], *RPL18A* [36], *PF4* [37,38], *PRG3* [39]) and T cell acute lymphoblastic leukemia (*TLX3* [40]). Moreover, some of these genes have already been reported to play a role in CML (*AURKB* [41], *AZU1* [42,43], *HLA-B* [44], *HLA-DMB* [30], *PF4* [45]). In addition, a subset of these genes has been associated with cancer therapy responses (*AZU1* [42,43], *CDCA3* [26], *CEACAM6* [46], *EN1* [28], *PF4* [37,38,45]). Thus, the characteristic phase-specific expression behavior and the reported functions of the genes with increased network connectivity may also contribute to the manifestation of differences between the three CML phases and potentially influence the expression of other signature genes.

### 2.6. Similarities and Differences of Imatinib-Resistant Patients to CML Phases

The initial study by [17] also measured the transcriptomes of 15 CML patients that were insensitive to the targeted therapy with imatinib. This enabled us to analyze how similar they are in relation to the three CML phases. Pairwise correlations between each imatinib-resistant patient and each patient of the three CML phases showed that the gene expression profiles of imatinib-resistant patients were most similar to those of patients in blast crisis, whereas the correlations to the accelerated or chronic phase were clearly smaller (Figure 6a). This is in good accordance with [17] which reported that transcriptomes of imatinib-resistant patients were similar to the advanced disease phase. Still, the observed differences between accelerated and blast phase again support that they represent distinct disease phases. Further, the strengths of the observed correlations with strongest values slightly greater than 0.6 indicated that relevant expression differences between imatinib-resistant patients and the CML phases may exist. Several hundred genes with altered expression that distinguished imatinib-resistant patients from the three CML phases were revealed by a comparative transcriptome analysis (Appendix A, *q*-value ≤ 0.05 and average absolute expression change |log${}_{2}$-ratio| ≥ 1: resistant vs. chronic: 820 under- and 1004 overexpressed genes, resistant vs. accelerated: 717 under- vs. 819 overexpressed genes, resistant vs. blast: 615 under- vs. 878 overexpressed genes). This included, for example, known tumor suppressor genes (e.g., *DACH1*, *MGAT2*, *SENP6*, *SMAD4*) that were jointly underexpressed or known cancer census genes (e.g., *BCL11B*, *CD74*, *GATA3*, *MAFB*, *PDGFB*) that were jointly overexpressed in all three comparisons of imatinib-resistant patients to the individual CML phases. Moreover, the altered genes were consistently enriched for overexpressed cytokine and ECM receptor interaction pathway genes for each comparison of the imatinib-resistant patients to one of the three CML phases (Figure 6b, Appendix A). This could potentially contribute to identify imatinib-resistant patients before treatment start and may help to develop other targeted treatment strategies for them.

## 3. Discussion

Nowadays, CML can be controlled efficiently for the majority of patients with targeted tyrosine kinase inhibitor therapies [1,5,6,8,9]. Fatal progressions from the chronic phase to the blast crisis are only rarely observed [6,10]. Still, in some cases treatment failures can lead to CML progression [8]. In addition, in cases when CML is first diagnosed in blast crisis [11,12,13], existing tyrosine kinase inhibitor therapies are often much less efficient [14,15,16]. Therefore, a gain in detailed knowledge of molecular alterations that distinguish the different CML phases is still very important. Such knowledge can contribute to the identification of underlying pathomechanisms and could provide a basis for the development of improved treatment strategies for advanced CML states.

To contribute to this, we performed an in-depth reanalysis of publicly available expression profiles of CML patients from [17] to characterize similarities and differences between the three CML phases with the help of well-established and recently developed bioinformatics approaches. In accordance with [17], we observed an increase of transcriptomic alterations in advanced CML states. The global expression differences were smaller between the accelerated and the chronic phase than between the accelerated and the blast phase. Most differentially expressed genes were observed for the comparison of the blast crisis to the chronic phase. In addition to [17], we performed a principal component analysis of the CML gene expression profiles and found that profiles of the chronic phase were more heterogeneous than those of the blast crisis, whereas accelerated phase samples were located between these two phases. This indicates that the chronological order of the phenotypically described phases of the progressive CML development [3] is also likely to be reflected at the transcriptomic level of phase-specific samples from different patients.

In contrast to [17], where it was suggested to consider the progressive development of CML as a two-step process from the chronic phase to an advanced phase (accelerated and blast), our analysis provides multiple lines of evidence which clearly indicate that one can consider the accelerated phase as a transition phase with its own characteristic expression profile between the chronic phase and the blast crisis. First of all, our comparative transcriptome analysis revealed several hundred genes that showed strong expression differences between the accelerated and the chronic phase and between the accelerated and the blast phase (Appendix A, *q*-value ≤ 0.05 and average absolute expression change |log${}_{2}$-ratio| ≥ 1: accelerated vs. chronic: 111 under- and 312 overexpressed genes, blast vs. accelerated: 664 under- vs. 416 overexpressed genes). Many of these genes were transcription factors and several of them were known cancer genes (e.g., *APC*, *BCL2*, *ERBB2*, *HRAS*, *TGFBR2*). Further, the heatmap clustering clearly separated the majority of accelerated phase samples from the majority of blast phase samples. Moreover, the accelerated phase differed from both other phases also at the signaling pathway level especially including significant alterations of the MAPK, adherens junction and cytokine receptor interaction pathway.

We also performed additional analyses that have not been performed with the data set from [17] before. Our analysis of differentially expressed genes in the context of known cancer-relevant signaling pathways revealed that the global pathway expression patterns had similar shapes across the CML phases. These patterns mainly differed by the number of altered genes that increased with the phase. Still, significant enrichments of altered pathway genes were observed for MAPK, PI3K-Akt and VEGF signaling and cytokine receptor interactions in dependency of the specific comparison of two CML phases. Several of these pathway alterations can be triggered by the BCR-ABL1 oncokinase [59], but also antagonistic relationships to BCR-ABL1 signaling are possible [60]. The crosstalk between these pathways could contribute to their global deregulation [61], which could, for example, influence the control of cell proliferation, differentiation, migration, adhesion and apoptosis with respect to the specific functions of these pathways. Thus, our global analysis indicates which signaling pathway alterations may play an important role in CML progression.

Moreover, we also considered the data set by [17] to learn CML-specific gene regulatory networks that were associated with the observed expression differences between the three CML phases. These networks also included 24 genes with phase-specific expression behavior that had an increased connectivity to other differentially expressed genes (Table 1). These genes could therefore be more important than others to distinguish between the three CML phases. This is further supported by the results of our in-depth literature analysis suggesting that (i) an accumulation of additional mutations (*NDUFAB1*, *SPRR2A*, *AURKB*), (ii) alterations of immune responses and inflammation (*HLA-B*, *HLA-DMB*, *HLA-DRA*, *MUC8*, *OPTN*), (iii) a global deregulation of signaling pathways (*CDCA3*, *EN1*, *RPL18A*, *ADD2*, *TMEM40*, *CEACAM6*), and (iv) an impact on treatment response and survival (*PF4*, *AZU1*, *CDCA3*, *EN1*) could be influenced by several of these genes. A detailed discussion of these genes in the context of the existing literature and the observed phase-specific expression behavior is provided in Appendix A. Moreover, two important drivers of other leukemias, *PRG3* and *TLX3*, were also among the 24 genes. *PRG3* encodes a proteoglykane that has been reported as major regulator associated with survival of acute myeloid leukemia patients [39]. *TLX3* encodes a homeobox transcription factor that is known as a master regulator of T cell acute lymphoblastic leukemia [40]. All these findings indicate the value of the network-based approach to identify genes that may play an important role in CML development and progression.

Further, it is important to discuss the value of our study in times when CML can be controlled by targeted tyrosine kinase inhibitor therapies for the majority of patients. Detailed knowledge of phase-wise gene expression alterations at the level of single genes and pathways contributes to a better understanding of CML development, may allow to identify patients that have an increased risk for progression, and could also provide hints for the development of improved treatment strategies for advanced phases. For example, the predicted moderate overexpression of *BCL2* in accelerated and blast phase suggests that drugs like venetoclax could potentially be an option for CML patients in advanced phases [62,63]. In addition, our in-depth comparison of imatinib-resistant patients to the three different CML phases predicted that increased activities of the cytokine and ECM receptor interaction pathways were associated with resistance to imatinib treatment. This finding could potentially be used to predict imatinib-insensitive patients before treatment start and may further offer the possibility to develop targeted treatments for them.

## 4. Materials and Methods

### 4.1. CML Gene Expression Data

Raw gene expression data files from [17] were downloaded from Gene Expression Omnibus (GSE4170). We considered all CML samples of the three CML phases (chronic phase: 42, accelerated phase: 9 defined by blast count and 8 defined by additional cytogenetic alterations, blast crisis: 28), which had been part of the main analysis by [17]. Note that the samples were from different patients and they do not represent a longitudinal follow-up study of individual patients across the phases. Further, we also considered the 15 imatinib-resistant patients from [17] for an additional comparative analysis in relation to patients of the three CML phases. The raw gene expression measurement of a gene in a sample was provided as log${}_{10}$-ratio of the hybridization intensity of this gene in the specific sample in relation to the corresponding hybridization intensity of this gene in a reference pool of chronic phase samples. We converted these raw expression values to log${}_{2}$-ratios and additionally performed quantile normalization of these raw gene expression measurements to standardize the gene expression values of the different samples to ensure a good comparability [64]. Normalized expression data of the 14,914 considered genes are provided in Appendix A for all samples. Principal component analysis of all CML samples was done using the R function prcomp.

### 4.2. Identification of Differentially Expressed Genes

Differentially expressed genes between each pair of the three different CML phases (accelerated vs. chronic, accelerated vs. blast, blast vs. chronic) were determined using the standard work flow of the R package limma [65]. Correction for multiple testing was done by computing false discovery rates using the method by Storey [66] implemented in the R package *q*-value. Results of the differential gene expression analysis are provided in Appendix A. Note that we initially performed a differential gene expression analysis to compare the samples of the accelerated phase defined based on histology to those defined based on additional cytogenetic changes. We only observed two differentially expressed genes at the *q*-value cutoff of 0.05 and therefore decided to treat both types of accelerated samples as one group. The heatmap of the CML gene expression signature was generated with the R function heatmap.2 using 1−r with *r* denoting the Pearson correlation as distance measure in combination with Ward’s hierarchical clustering method (ward.D2) [67]. Further, the same basic limma framework was used to determine differentially expressed genes between imatinib-resistant patients and patients of each CML phase (Appendix A).

### 4.3. Gene Annotation Analysis

Cancer-relevant gene and signaling pathway annotations were taken from [68]. Under- and overexpressed genes were determined separately for each annotation category. Enrichment of genes in a specific annotation category was tested using Fisher’s exact test. Correction for multiple testing was done by computing FDR-adjusted *p*-values using the R function p.adjust.

### 4.4. Inference of Signature-Specific Gene Regulatory Networks

All 9884 differentially expressed genes of the gene-signature that distinguished the three CML phases (*q*-value ≤ 0.05) were considered to learn a signature-specific gene regulatory network (Appendix A). The expression of each signature gene was modeled as a linear combination of the weighted expression values of all other signature genes. The parameters of the underlying linear models were determined using the R package regNet [24], which uses lasso regression [69] in combination with a significance test for lasso [70] to determine the most relevant predictors for each gene-specific linear model. Depending on the sign of the learned weight parameter, a selected predictor can either represent a potential activator (positive weight) or inhibitor (negative weight) of a specific signature gene. Since each signature gene can be selected as a potential regulator of other genes, a global signature-specific network is fully determined by the signature gene-specific linear models. More details to the underlying concept are provided in [24,68]. Following [71], this network inference was repeated 100 times based on randomly created training sets that comprised 75% of all 87 CML gene expression samples (i.e., 65 randomly selected CML samples). The other 25% of samples (i.e., the remaining 22 CML samples), which were not in the corresponding specific training set, were considered as independent network-specific test set. The prediction quality of each learned network was determined based on its corresponding test set by computing the correlation between predicted and originally measured gene expression levels. These computations were done for each originally learned network including all network links at a *q*-value cutoff of 0.01 and for its 10 corresponding random network instances of same complexity derived by degree-preserving network permutations. The prediction quality of each gene was further averaged across the networks to compare the predictive power of the originally learned networks to those of the corresponding random networks. The gene-specific correlations observed for the learned networks were significantly greater than zero (Wilcoxon signed-rank test: *p* < $2.2\times 10^{-16}$, median correlation: 0.63) and also significantly greater than those reached by corresponding random networks of same complexity (U-test: *p* < $2.2\times 10^{-16}$, median correlations: 0.63 vs. 0.01) demonstrating that the learned networks contained relevant information for the prediction of CML gene expression levels (Appendix A). The majority of signature genes had one outgoing link to another signature gene and only relatively few genes with more than two outgoing links to other signature genes were observed (Appendix A: red curve). There were also more than 300 signature genes that did not have an incoming link from another signature gene (Appendix A: blue curve). To obtain an integrative view of gene expression differences between the different CML phases, we used the R package igraph (Kamada–Kawai layout algorithm) to visualize a consensus network. This was done based on links that were present in at least 90 of 100 networks at a *q*-value cutoff of 0.01 (Appendix A) to display genes with at least two outgoing edges.

## 5. Conclusions

We performed an in-depth bioinformatics analysis of publicly available gene expression profiles from CML patients of different disease phases. Our comparative analysis of the three CML phases revealed similarities and differences at the level of single genes, signaling pathways and gene regulatory networks. The derived gene expression signature highlighted characteristic gene expression differences between all three CML phases suggesting that one can consider CML development from the chronic phase to the blast crisis as a three- rather than a two-step process. Global expression alteration patterns of signaling pathways were very similar for the three CML phases but the number of affected genes increased from the chronic to the blast phase. Especially the cytokine receptor interaction pathway, which was significantly altered from phase to phase when focusing on genes with stronger expression differences, might play an important role in the phase-wise development of CML. Further, our in-depth literature analysis of the network-based predicted potential major regulators with their phase-specific expression profiles revealed that several of these genes could contribute to an accumulation of additional mutations, alterations of immune responses, a global deregulation of signaling pathways or have the potential to influence treatment response and survival. Thus, our study contributes to a better understanding of molecular alterations that distinguish the three CML phases. Such knowledge is still very important and may help to develop new treatment strategies for patients that do not profit from existing highly efficient tyrosine kinase inhibitor therapies. In accordance with this, the moderate increase of the expression of *BCL2* in the accelerated and blast phase suggests that existing drugs such as venetoclax could represent a potential treatment option for advanced phase patients. Further, the observed characteristic overexpression of cytokine and ECM receptor interaction pathway genes between matinib-resistant patients and each CML phase could help to predict patients that may not profit from an imatinib therapy and could potentially contribute to develop new treatment strategies for them.

## Figures and Tables

**Figure 1 cancers-14-00256-f001:**
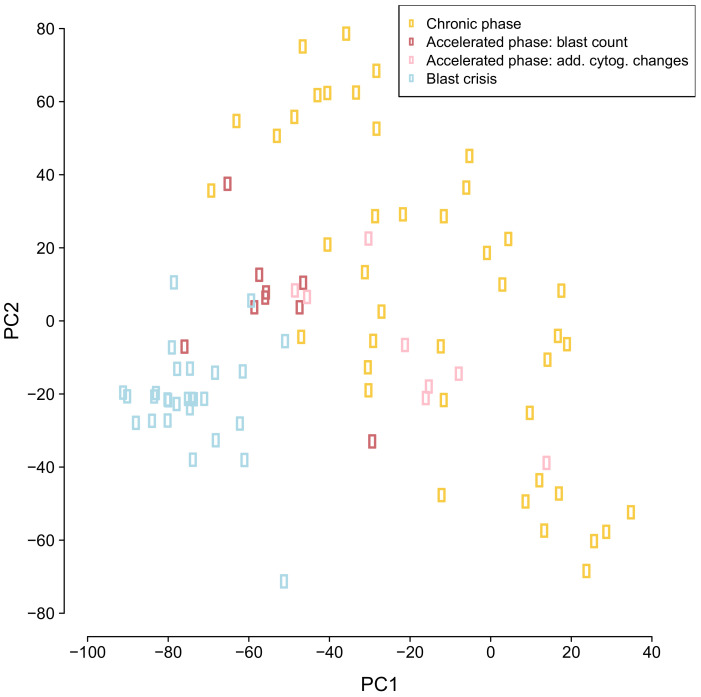
Principal component analysis of genome-wide CML expression data. Individual patient samples are represented by colored dots. The color of a dot defines the CML phase of the sample (yellow: chronic phase, red/pink: accelerated phase, light blue: blast crisis).

**Figure 2 cancers-14-00256-f002:**
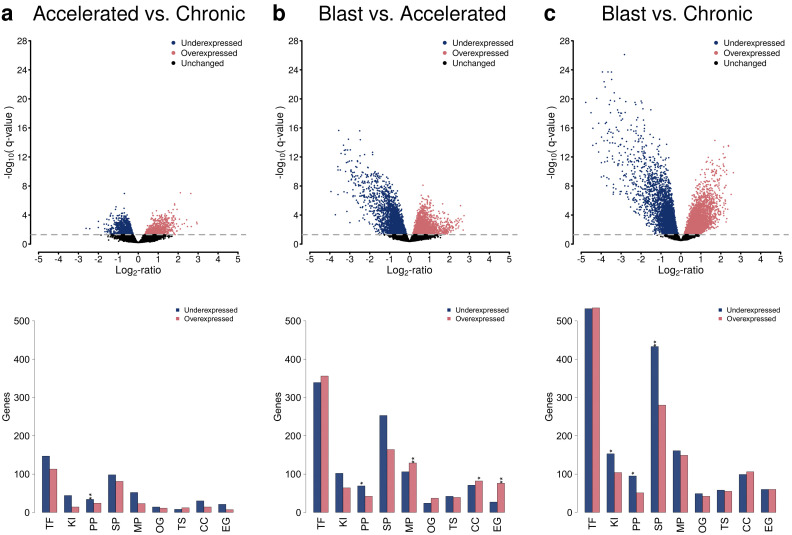
Differential gene expression analysis and functional annotation analysis of altered genes based on pairwise comparisons of the three CML phases. Volcano plots quantifying the average expression changes of genes (*x*-axis) in relation to the obtained significance of gene expression difference (*y*-axis) are shown in the first row. The gray dotted line represents the significance cutoff (*q*-value: 0.05). Underexpressed genes are represented by blue dots and overexpressed genes by red dots. The corresponding numbers of under- and overexpressed genes in different functional annotation categories (TF: transcription factor or co-factor, KI: kinase, PP: phosphatase, SP: signaling pathway gene, MP: metabolic pathway gene, OG: oncogene, TS: tumor suppressor gene, CC: cancer census gene, EG: essential gene) are shown in the bar plots in the second row. Enriched genes in a specific functional category are highlighted by asterisks (Fisher’s exact test: ‘**’: FDR-adjusted *p* ≤ 0.01, ‘*’: FDR-adjusted *p* ≤ 0.05). The comparisons of accelerated to chronic phase, blast to accelerated phase, and blast to chronic phase are shown in the columns (**a**–**c**).

**Figure 3 cancers-14-00256-f003:**
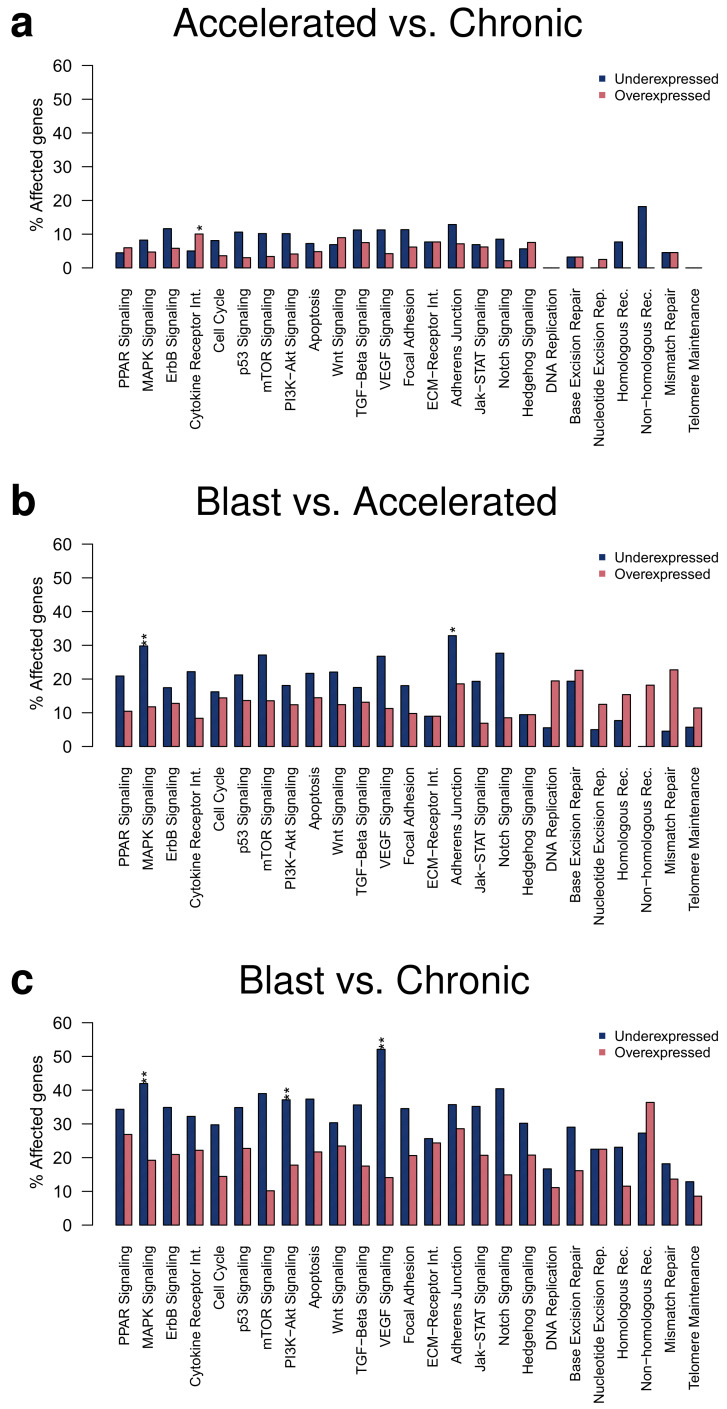
Gene expression alterations affecting cancer-relevant signaling pathways. Percentages of under- and overexpressed genes observed for specific signaling pathways are shown for the pairwise comparisons of the three CML phases. Differentially expressed genes at the *q*-value cutoff of 0.05 were considered and overrepresented pathways were marked by asterisks separately for an enrichment of under- or overexpressed genes (Fisher’s exact test: ‘**’: FDR-adjusted *p* ≤ 0.01, ‘*’: FDR-adjusted *p* ≤ 0.05). Signaling pathway alterations for the comparisons of accelerated to chronic phase, blast to accelerated phase, and blast to chronic phase are shown in the subpanels (**a**–**c**), respectively.

**Figure 4 cancers-14-00256-f004:**
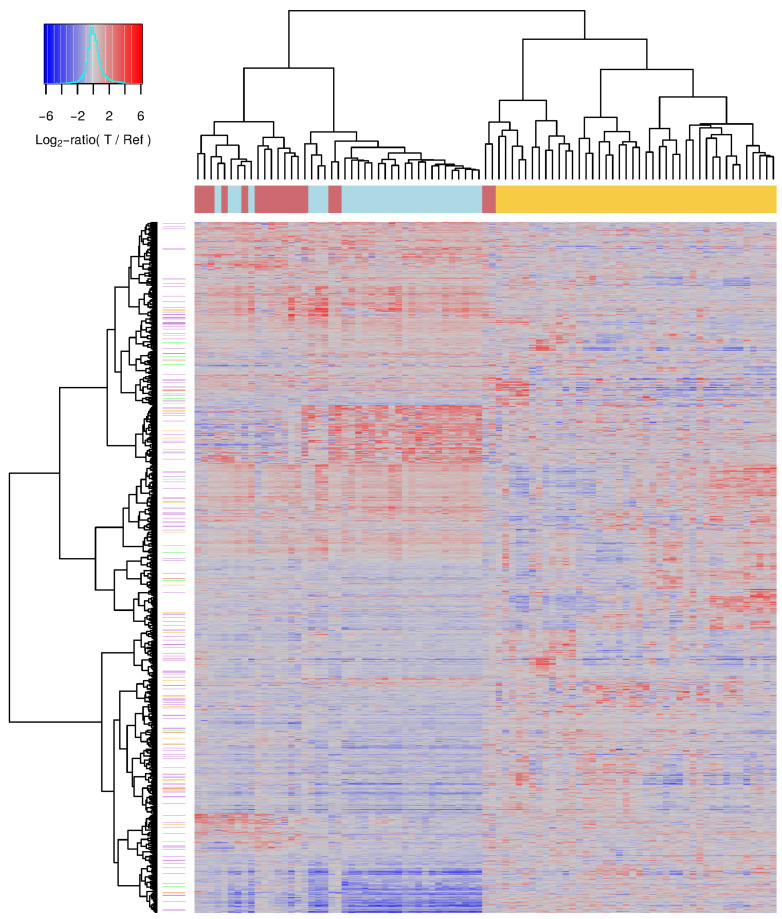
Differentially expressed genes distinguishing the three CML phases. Heatmap of individual CML gene expression profiles. CML samples were clustered based on the similarity of their gene expression profiles (columns) and their corresponding gene-specific expression values are visualized (rows). All differentially expressed genes of the pairwise comparisons of the three CML phases at the level of the *q*-value cutoff of 0.05 are included. The heatmap represents log${}_{2}$-ratio expression values of each specific CML sample in comparison to a pool of reference samples visualizing underexpressed genes with values clearly less than zero in blue shades, unchanged genes with values about zero in gray, and overexpressed genes with values clearly greater than zero in red shades of the specific sample. The colored line below the column dendrogram at top of the heatmap represents the phase-specific label of each sample (yellow: chronic phase, red: accelerated phase, light blue: blast crisis). Lilac, orange and green lines right to the row dendrogram of the heatmap mark known signaling pathway genes, cancer census genes and genes that are part of both categories, respectively.

**Figure 5 cancers-14-00256-f005:**
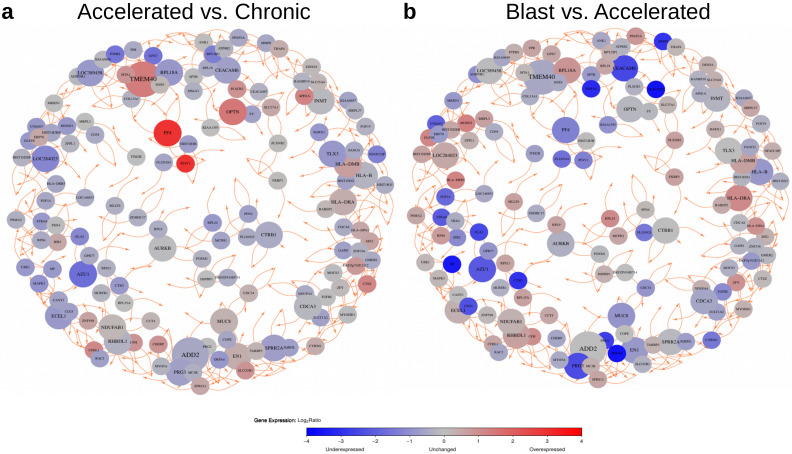
Network-based visualization of genes with increased network connectivity and corresponding expression behavior for pairwise comparisons of CML phases. Nodes represent selected genes with at least two outgoing links to other genes that were present in at least 90 of the 100 learned networks. Node sizes are proportional to their out-degrees and nodes are colored according to expression change observed for the pairwise comparison of CML phases (blue: underexpressed, gray: unchanged, red: overexpressed). Predicted activating links between genes are shown in orange. Links can represent direct or indirect regulatory dependencies or may only represent correlations between expression levels of genes. The network (**a**) shows expression alterations of the selected genes with increased connectivity to other signature genes for the comparison of the accelerated to the chronic phase. The network (**b**) shows the expression behavior of these genes for the comparison of the blast to the accelerated phase.

**Figure 6 cancers-14-00256-f006:**
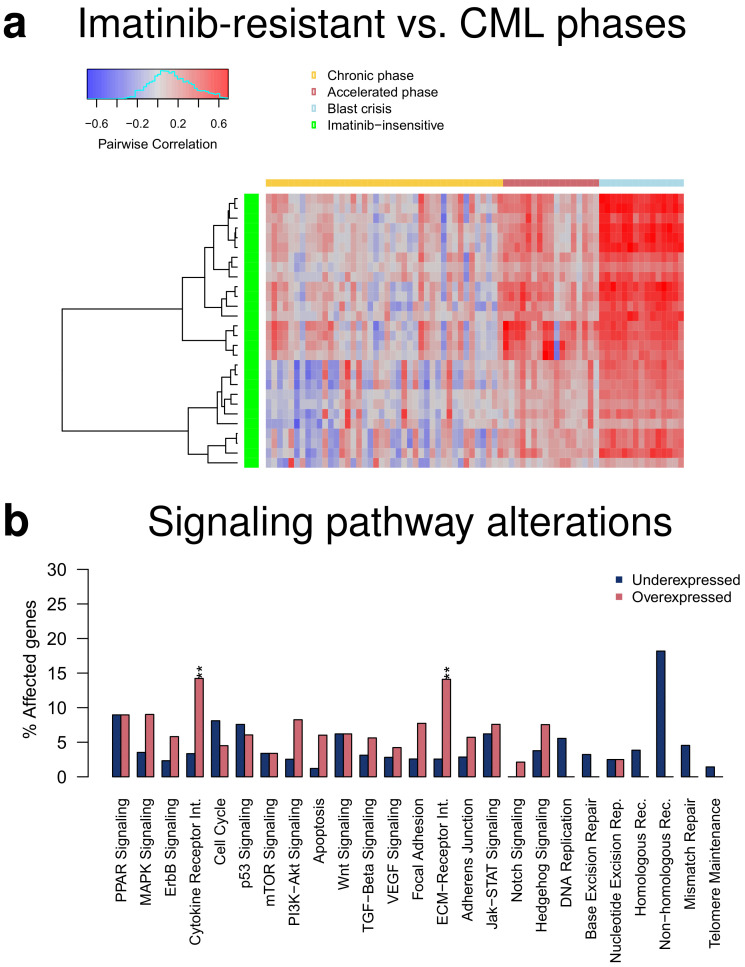
Comparison of transcriptomes of imatinib-resistant patients to patients of the three CML phases. (**a**) Heatmap of pairwise correlations between genome-wide gene expression profiles of imatinib-resistant patients and individual patients of the three CML phases. The imatinib-resistant patients are shown in the rows of the heatmap clustered according to the similarity of their correlation profiles (green: imatinib-resistant). The individual patients from the three CML phases are shown in the columns of the heatmap in phase-wise order (yellow: chronic phase, red: accelerated phase, light blue: blast crisis). The heatmap represents negative correlations in blue shades, correlations about zero in gray shades, and positive correlations in red shades. (**b**) Bar plot of percentages of under- and overexpressed genes of specific signaling pathways comparing imatinib-resistant patients to patients in blast crisis. Differentially expressed genes at the *q*-value cutoff of 0.05 with an average absolute expression change of |log${}_{2}$-ratio| ≥ 1 were considered. Overrepresented pathways are marked by asterisks for an enrichment of under- or overexpressed genes (Fisher’s exact test: ‘**’: FDR-adjusted *p* < 0.01). Identical enrichment patterns were also observed for the comparison to the chronic and accelerated phase (Appendix A).

**Table 1 cancers-14-00256-t001:** Genes with increased network connectivity and corresponding expression behavior for pairwise comparisons of CML phases. Listed genes had at least three outgoing links to other signature genes in the signature-specific network in Figure 5. These genes were either significantly under- (’-’) or overexpressed (’+’) or they were unchanged (’=’) in pairwise comparisons of accelerated to chronic phase (AP vs. CP), blast crisis to accelerated phase (BC vs. AP), or blast crisis to chronic phase (BC vs. CP) at a *q*-value cutoff of 0.05 (Appendix A). Functional annotations related to cancer are listed and references to cancer-relevant or other publications are provided.

Gene	AP vs. CP	BC vs. AP	BC vs. CP	Selected Functional Annotations	References
* ADD2*	=	=	-	migration, proliferation	[47,48]
* CDCA3*	=	=	-	proliferation, cisplatin sensitivity	[26,35,49]
* CTRB1*	=	=	-	serine protease	[25]
* ECEL1*	=	=	-	regulation of hormones and neuropeptides	[27]
* AURKB*	=	-	-	chromatid segregation	[41,50]
* CEACAM6*	=	-	-	adhesion, proliferation, apoptosis, differentiation,	[46]
				invasion, metastasis, therapy response	
* HLA-B*	=	-	-	immune response	[44]
* INMT*	=	-	-	enzyme, methyltransferase	[25]
* PRG3*	=	-	-	proteoglykane, survival	[39]
* AZU1*	-	-	-	therapy response and survival	[42,43]
* OPTN*	=	=	+	inflammation, apoptosis	[51]
* HLA-DMB*	=	=	+	immune response, survival	[29,30]
* NDUFAB1*	=	=	+	oxidative stress, gain of mutations	[52]
* HLA-DRA*	=	+	+	immune response	[25]
* EN1*	=	-	=	homeobox gene, differentiation, therapy response	[28,53]
* MUC8*	=	-	=	anti-inflammation	[54]
* LOC389458*	-	=	-	uncharaterized	
* SPRR2A*	-	=	-	local invasiveness, protection oxidative stress	[31]
* TLX3*	-	=	-	homeobox gene, driver of T-ALL	[40]
* LOC284023*	-	+	-	uncharaterized	
* RHBDL1*	=	+	=	potential intramembrane serine protease	[25,55]
* RPL18A*	=	+	=	part 60S ribosomal subunit, lack red blood cells	[36,56]
* PF4*	+	=	+	regulation hematopoietic stem and progenitor cells	[37,38,45,57,58]
* TMEM40*	+	=	+	apoptosis, proliferation, migration, invasion	[32,33,34]

## Data Availability

Raw gene expression profiles from [17] are available from Gene Expression Omnibus (GSE4170). Processed gene expression profiles considered in our study are available in Appendix A.

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
