# Peer review of "Comparative Gene Expression Analysis Reveals Similarities and Differences of Chronic Myeloid Leukemia Phases"

_cancers, 2022, doi:10.3390/cancers14010256_

Round 1

Reviewer 1 Report

A comprehensive paper that identified potential biomarkers for CML.

Before publication write the Genes abbreviations with Italic.

Author Response

We thank you for the positive feedback to our study. Please consider the section Reviewer #1 in the attached PDF file below to see how we addressed your comments in the revised manuscript.

Reviewer 2 Report

This study is a in-depth bioinformatics analysis of publically available gene expression profiles from patients diagnosed in one of the three phases of CML: chronic, accelerated and blast phase disease.  The authors did pairwise comparisons that showed there was a different gene expression profile for each phase of CML and that the evolution from chronic to blast phase disease is a three step process. The aim of the study was really to differentiate between the three phases of CML using analysis of publically available data.   Significant alterations in specific genes/pathways characterized each phase of CML. In this public database, however, the catagorization of each phase may not be completely correct but there was little overlap between the three phases. The heterogenity of the disease was greatest in chronic and accelerated phase and less so in blast phase of the disease.  

The content is interesting and the view of CML as a three step process had been controversial.  Radich et al had done molecular analysis suggesting that the evolution from chronic-->accelerated-->blast phase was really a two step process. This paper presents strong evidence that there are specific gene  expression profiles unique to each phase of CML. One problem with the paper is that it is aimed at readers who have a good knowledge of bioinformatics methods and can best determine the validity of their findings. It is a difficult read without more clarification of methods for the general hematologist interested in CML.  A better explanation of the methods would allow for a better understanding of the results.  I am not an expert on the various bioinformatics techniques which made it difficult to fully appreciate the results.  There are many distinct genes and pathways that are effected but these are not presented in an optimally organized manner.  The figures were helpful and certainly were convincing that there are three distinct phases of CML progression.  

What is the value of this data clinically when the majority of patients with CML have disease control with TKI therapy?  Understanding the gene expression profiles of each phase may allow for the identification of patients who are more likely to progress to accelerated or blast phase or who may not respond optimally to TKI treatment.  This kind of analysis may identify patients more likely to undergo clonal evolution.  It also may assist in the identification of patients who are likely or unlikely to respond to a trial of TFR.  In patients that do progress their disease, this type of analysis may suggest possible drugs to target accelerated and blast phase CML.  Overexpression of BCL2, for example, suggests that drugs like venetoclax might be valuable in more advanced phases of CML. 

It would be a more clinically useful analysis to look at the changes that occur in patients who are not optimal responders to TKI therapy and look at the trajectory of their gene expression. For example  those that have a suboptimal response to TKI therapy after initiation of treatment, those who have optimal response but are unable to succeed at trial of TFR, and those that progress their disease to accelerated and blast phase.  That needs to be the aim of studies such as this one in order to make a difference in CML treatment outcomes.

Stylistically, the paper is not written in a clear manner.  The figures allow the reader to figure out the story but the language does not present the data in an organized manner.  It needs a rewrite with a more economical style that gets the major points outlining the reasons for the study, the results, why is refutes the data suggesting two stage evolution of leukemia progression, and how this data can be used (next steps) for improving CML outcome. 

Author Response

We thank you for the very good summary, positive feedback and comments that helped us to further improve the manuscript. Please consider the section Reviewer #2 in the attached PDF file below to see how we addressed your comments in the revised manuscript.

Reviewer 3 Report

In this study authors have reanalysed Chronic myeloid leukemia (CML) samples for a more detailed characterization of molecular alterations that distinguish the different CML phases. They performed an in-depth bioinformatics analysis of publicly available gene expression profiles of the three CML phases. Reported characteristic gene expression signature which clearly distinguished the three CML phases. Similar signaling pathway were identified in

among the three phases but differed strongly in the number of affected genes, which increased with the phase. Significant alterations of MAPK, VEGF, PI3K-Akt, adherens junction and cytokine receptor interaction signaling were marker of specific phases. Also suggested the phase-wise CML development as a three rather process in accordance with which phase-specific expression behaviour of 24 potential major regulators were predicted. Out of which some of these genes have already been reported in relation to CML (e.g., AURKB, AZU1, HLA-B, HLA-DMB, PF4) and others have been found to play important roles in different leukemias (e.g., CDCA3, RPL18A, PRG3, TLX3).

Comments

Authors comparative re-analysis of only one study may not strongly contributes to an in-depth molecular characterization of similarities and differences of the CML phases. More studies should be included for better outcome like phase wise CML analysis has also been conducted in Oncotarget, 2018, Vol. 9, (No. 54), pp: 30385-30418.

In my opinion a meta-analysis may provide more detailed characterization of molecular alterations and help us to better understand similarities and differences of the CML phases.  

Author Response

We thank you for the very good summary and the feedback to our study. Please consider the section Reviewer #3 in the attached PDF file below to see how we addressed your comments in the revised manuscript.

Round 2

Reviewer 3 Report

I agree with authors to retain their basic design study both because of nonavailability of data as well as the gene expression profiles in the study by Singh et al. 2018 were measured on a different platform than authors study. In addition, Singh et al. 2018 mainly focused on the comparison of TKI-resistant to TKI-sensitive patients. However, similarities and differences of Imatinib-resistant patients to three CML phases in the present study has brought more clarity and has improved the study.